# Reporting maternal deaths in Anglophone West Africa: A media content analysis of articles published online between 2015 and 2025

Kiran Roy[1], Noimot Balogun[2,3], Aduragbemi Banke-Thomas[1,2]*

1 Faculty of Epidemiology and Population Health, London School of Hygiene and Tropical Medicine, London, United Kingdom, 2 Maternal and Reproductive Health Collective, Lagos, Nigeria, 3 Faculty of Communication and Media Studies, University of Lagos, Lagos, Nigeria

* aduragbemi.banke-thomas@lshtm.ac.uk

## Abstract

Maternal mortality is a serious public health issue globally, with countries in West Africa facing some of the highest burdens. Media reportage of maternal deaths can have a significant influence on public health policy. We conducted a content analysis of digital media across the five Anglophone West African countries (The Gambia, Ghana, Liberia, Nigeria, and Sierra Leone) to understand what is reported about maternal deaths and how they are reported. Four to five widely read online newspapers and two of the most popular blogging sites from each country were selected. For each source, we searched for relevant articles, retrieving and including those with a detailed report of maternal death due to obstetric causes. Following data extraction, we used quantitative and qualitative analyses, using the three-delay model and a derived standards checklist based on the Principles of the Association of Health Care Journalists and the Impress Standards Code for coding, respectively. Inter-coder reliability was assessed for quantitative analysis, and audit trail and debriefing conducted for qualitative analysis. Fifty-three detailed articles, consisting of 35 online newspaper articles (61%) and 16 blogs (30%), were included. Most were published in 2023 (30%) and in Nigeria (75%). Delays in facilities, including negligence, malpractice, long waiting times, and withholding care contingent on payment, were frequently mentioned. All articles distinguished fact from opinion, and authors avoided taking a side, 90% were respectful, 68% captured >1 perspective, with most capturing perspectives of the spouse and the government; however, only 6% consulted independent experts. Overall, media reportage of maternal deaths in Anglophone West Africa confirms existing and offers new insights. Preservation of the dignity and respect of pregnant women in death; engagement of multiple voices, including independent experts, and inclusion of actions being undertaken or required to prevent future occurrences will improve reporting and its utility for policy change.

**Data availability statement:** All relevant data are within the paper and its Supporting information files.

**Funding:** The authors received no specific funding for this work.

**Competing interests:** The authors have declared that no competing interests exist.

## Introduction

Maternal mortality, defined as any death that occurs during pregnancy or within 42 days of termination of pregnancy, is a serious public health issue that leads to 260,000 preventable deaths per year. These deaths are usually due to obstetric causes such as bleeding, hypertension, infection, or abortion. While there has been a 34% reduction in maternal mortality between 2000 and 2023 [1], at the current rate of decline, the Sustainable Development Goal (SDG) target set in 2015 to reduce the global maternal mortality ratio (MMR) to 70 deaths per 100,000 live births by 2030 is unlikely to be achieved [2]. Currently, despite sub-Saharan Africa (SSA) making up only 15% of the world's population [3], it is where 70% of all reported maternal deaths globally occurred (i.e., 187,000). West Africa alone, comprising eight Francophone, five Anglophone, and two Lusophone countries, contributes over 50% (i.e., > 100,000) of the maternal deaths reported within SSA [1].

Beyond numerical enumeration of maternal deaths and the estimation of MMR, public health experts and policymakers recognise the importance of examining the underlying causes of maternal death and the circumstances in which the deaths occur. Such evidence helps to drive more targeted interventions that can contribute to tangible reductions in maternal mortality. For the most part, such evidence is captured and reported in verbal and social autopsies, which are typically deployed as part of maternal and perinatal death surveillance and review (MPDSR) conducted in communities. Verbal autopsy is a method used "to determine the medical cause of death through interviews with the deceased person's next of kin or caregivers" [4]. Social autopsy extends beyond the medical causes of death by incorporating socio-cultural and behavioural questions [5,6]. Both methods help improve understanding of the causes of maternal deaths; however, MPDSR implementation in many West African countries has been fraught with numerous challenges, including the cost associated with routine data collection, which makes it difficult for governments to sustain [7–9].

The media, encompassing both traditional (e.g., newspapers, television, and radio) and new/digital media (e.g., online newspapers, blogs, online reports, etc.), is another source of information on health topics, including maternal deaths [10,11]. The media plays a crucial role in shaping the public's understanding and perception of public health issues, with digital media deemed particularly effective in this regard [12–14]. Beyond informing the public, the media exerts significant influence on real-world events by holding power to account, amplifying marginalised voices, and guiding policymakers' decisions. In the area of maternal health, the media has been used to convey the severity of the maternal mortality crisis, the impact of implemented interventions, and the need for improved health policies and legislation [15].

Given the high burden of maternal mortality and the persistent challenges in implementing community-based MPDSR in West Africa [1,9], it is important to explore the potential of alternative data sources for understanding causes of maternal deaths in the region. Data from the media offers a viable alternative, particularly since it imposes no direct cost on the government. Yet, to date, no study reviewing media content on maternal deaths in West Africa has been published to date. Analysis

attempting to address this knowledge gap must control for reporting language, recognising that distinct journalistic traditions and media systems are inextricably linked to linguistic heritage and context [16,17]. With recognition that language differentially shapes how public health issues are communicated in the media [18] and approximately four of five maternal deaths in West Africa occurring in the region's Anglophone countries [1], our study aimed to conduct a media content analysis to examine what about and how maternal deaths are being reported in digital media in Anglophone West Africa.

## Methods

### Study design

We used a media content analysis, which can be used to gain insights into the messages and images in discourse and popular culture, to consider how the media reflects society, and to examine the potential effects mass media could have on its audience [19]. This study design has similarly been used to examine newspaper coverage of progress made towards reducing maternal mortality in two sub-Saharan African countries, Rwanda and South Africa, previously [15]. In our study, we closely followed guidance and best practices for conducting media content analysis, as outlined by Macnamara (2005), using both quantitative and qualitative approaches to gain a full picture of the meaning and contextual messages of the reports [20].

### Study setting

Ghana, Liberia, Nigeria, Sierra Leone, and The Gambia are the five Anglophone West African countries and have all been selected for this study. Based on estimates by the World Health Organization (WHO), all five countries combined contributed 79,450 (79%) of the 100,230 maternal deaths reported in West Africa in the year 2023, a burden largely driven by Nigeria, which alone contributed 75,000 maternal deaths (Table 1). Francophone countries in the region contributed 20,450 (20%), and Lusophone countries contributed 330 maternal deaths (<1%) [1].

Based on the WHO MMR categorisation, Nigeria is classified as "Very High," with around 993 deaths per 100,000 live births. Liberia is classified as "Very High" with around 628 deaths per 100,000 live births. Gambia and Sierra Leone are both classified as "High," with 354 deaths per 100,000 live births. Lastly, Ghana is classified as "Moderate," with around 234 deaths per 100,000 live births (Table 1) [1].

According to a recent 2022 scoping review, various individual factors relating to delays in seeking care, health facility factors such as sub-optimal triage, monitoring, and referral, and wider health system factors such as transportation between health facilities contribute to maternal deaths across Anglophone West African countries [21]. Liberia experiences additional issues due to its 14 years of civil war, which, though ended in 2003, continues to influence maternal health service utilisation [22].

Table 1. Estimates of country-level epidemiological data in 2023[1.]

| Country | Maternal deaths per 100,000 live births | Maternal deaths | Lifetime risk of maternal death | WHO Classification |
| --- | --- | --- | --- | --- |
| Nigeria | 993 | 75000 | 1 in 25 | Very High |
| Liberia | 628 | 1100 | 1 in 40 | Very High |
| The Gambia | 354 | 290 | 1 in 72 | High |
| Sierra Leone | 354 | 920 | 1 in 74 | High |
| Ghana | 234 | 2100 | 1 in 133 | Moderate |
| Sub-Saharan Africa | 454 | 182000 | 1 in 55 | – |
| World | 197 | 260000 | 1 in 272 | – |

Footnote: Data in the table is based on the Trends in maternal mortality 2000–2023: Estimates by WHO, UNICEF, UNFPA, World Bank Group and UN-DESA/Population Division.

## News source selection

We selected English language sources only to standardise the analysis. To select online newspapers, four to five leading newspapers in each country were chosen based on their circulation numbers, which can indicate popularity. The circulation numbers were retrieved by querying ChatGPT (OpenAI, San Francisco, California, USA). The output was validated by asking colleagues who live or work in the countries whether the identified papers were the most widely read in each country. In the case where colleagues local to the country disagreed with the generated list, the proposition from local colleagues was prioritised. For our study, an online newspaper was the digital version of a traditional newspaper, accessible via the internet, which aims to deliver factual accounts of current events. For Nigeria, The Nation, Vanguard, Daily Trust, The Guardian, and The Punch were selected. For Liberia, the following were selected: Liberian Observer, The Analyst, FrontPage Africa, The New Dawn, and The Inquirer. For The Gambia, The Standard, The Point, Kairo News and Foroyaa were selected. Awoko Times, Cocorioko, Concord Times and Sierra Leone Telegraph were chosen for Sierra Leone. The Daily Graphic, Daily Guide, The Mirror, The Chronicle, and Daily Mail were selected for Ghana. In addition, we selected blogging sites, recognised in this study as non-formal, solely online platforms that share written content as blog posts. Specifically, we selected the most popular blogging sites that capture and report real societal issues and have the highest levels of internet traffic. As with online newspapers, these were identified by querying ChatGPT and validating with local colleagues. We selected *What's On Gambia* and *Gambiana* for The Gambia, *GhanaWeb News* for Ghana, *BellaNaija* and *Linda Ikeji's Blog* for Nigeria, AllAfrica News and *Front Page Africa* for Liberia, and *Sierraloaded* and *Swit Salone* for Sierra Leone. To find additional sources, a social media campaign was launched, featuring a poster shared across multiple social channels (LinkedIn, Twitter, and WhatsApp groups) by KR and ABT. After this campaign, additional sources, including blogging and investigative reporting sites, were identified (S1 File).

## Search strategy

An initial preliminary search was conducted through an iterative process to pilot test several search terms ("*maternal mortality*", "*maternal death*", "*dead woman*", "*pregnant woman*", "*pregnancy-related death*", "*pregnancy-associated death*", "*pregnancy complication*" and "*obstetric mortality*") and assess the relevance of articles retrieved to the study. The preliminary search was conducted using the search tab on the websites of various news sources. Following this, the search terms *"pregnant woman"*, "*maternal mortality*", and *"maternal death"* were assessed as most sensitive to identify relevant articles, with the presence of one or the other (*"pregnant woman"* OR "*maternal mortality*" OR *"maternal death"*) warranting an assessment of the identified article.

Each identified article was preliminarily screened for its content, then categorised based on its comprehensiveness and capacity to help us understand what is being reported about maternal deaths and how. Specifically, we grouped retrieved articles into one of five categories: (1) a detailed report of maternal death due to obstetric causes, (e.g., bleeding and high blood pressure, as defined by the WHO [1]) (2) a report about maternal death due to obstetric causes, but undetailed, (3) a report about a maternal death due to accidental/incidental causes, (e.g., road traffic accident, murder) (4) a report about a maternal death that almost occurred, but was averted, and (5) a report describing maternal mortality as an issue. Our preliminary screening of all retrieved articles revealed that articles with more than 250 words tended to have sufficient textual depth needed for meaningful content analysis, including the identification of themes, frames, and discursive patterns [20]. On the other hand, shorter reports typically lacked the narrative and contextual complexity required for a comprehensive understanding of the cause of the maternal death. As such, retrieved articles that were assessed as being Category 1 underwent further screening for eligibility into the study. A Preferred Reporting Items for Systematic Reviews and Meta-Analysis (PRISMA) diagram was used to annotate the search strategy [23]. The search was initially conducted between February and June 2025 and updated in January 2026.

## Eligibility criteria

For Category 1 articles to be included in this study, they needed to 1) describe a real event (maternal death) that occurred in The Gambia, Ghana, Liberia, Nigeria, or Sierra Leone, 2) contain an obstetric cause of the maternal death, as defined by the WHO [1], 3) provide a detailed narrative account of the death, with a minimum length of 250 words (the benchmark selected for sufficient textual depth). The narrative should describe the sequence of events leading up to the death and the circumstances surrounding it; and include verifiable identifying information—such as the location of death or the name of the deceased—unless the inclusion of such details is restricted due to privacy, ethical, or legal considerations, and 4) be published after 2015 being the start year of the SDGs. We included all articles that met these inclusion criteria, even if they reported the same maternal death, since articles could present the same death in different ways. Articles were excluded if they were undetailed reports (i.e., less than 250 words), described the death of a pregnant woman that was due to accidental/incidental causes, were follow-up stories of the same death from the same source, or there was concern about the authenticity of the report.

## Data extraction

All included articles were downloaded from the news sites and imported into NVivo (Lumivero, Denver, Colorado, USA). Articles were organised by country and news source. From all included articles, data on news source, name(s) of author(s), title, country, publication year, perspectives included, and hyperlink of the published article were extracted into a pre-tested data extraction Google Sheet (Google LLC, Mountain View, California, USA). Excerpts from the articles describing the cause of maternal death were extracted.

## Data analysis

We first described the total number of articles and the number per country and identified the number of unique stories of maternal deaths reported. A coding frame was developed and applied to analyse the content of the included articles and convert it to numerical data for analysis. The coding frame helped to establish: (1) delay phase(s) reported to contribute to the death (what about the death) and (2) how the maternal death was reported (S2 File).

We used the three-delay framework to map and assess what was reported in the retrieved articles regarding maternal deaths, including the events leading to the death and the factors contributing to it. The framework identifies three phases of delays that contribute to maternal deaths. Phase I delay describes delays in the decision to seek care, Phase II describes delays in reaching an adequate healthcare facility, and Phase III describes delays in receiving adequate care at the facility [24]. We conducted quantitative and qualitative content analyses using the three-delay framework. For the quantitative analysis, we mapped the events in the retrieved articles to the concomitant phases identified in the articles and reported the number of articles that contributed to maternal deaths in each phase. For the qualitative content analysis, we used a deductive approach to analyse the specific events deemed to have contributed to the death following familiarisation with the content. In addition, we described any advocacy points or recommendations for actions reported in the included articles.

We adapted professional standards for reporting, as established by the Association of Health Care Journalists (AHCJ)'s principles of health reporting [25,26] and the Impress Standards Code [27] to assess how maternal deaths are reported. In this study, we assessed included articles against 10 derived professional standards from both professional standards which could be solely assessed in content of reporting: 1) Recognise most stories involve degree of nuance, 2) Be accurate, 3) Show respect, 4) Avoid vague, sensational language, 5) Seek out independent experts, 6) Distinguish between advocacy and reporting, and 7) Be original (Table 2). For the quantitative content analysis, we assessed articles that met each of the 10 criteria, awarding 1 point if the criterion was fully met, 0.5 points if partially met, and 0 points if not met. Articles were classed as high quality if they scored ≥9, average quality if they scored ≥7 but <9, and poor quality if they

**Table 2. Criteria used for assessing how maternal deaths were reported.**

| S/No | Principles | Standards |
|------|-----------|-----------|
| 1) | Recognise most stories involve degree of nuance | 1a. Capture perspectives of various stakeholders implicated in or affected by the death |
| | | 1b. Have named or be able to name sources, if required, except if confidentiality has been agreed and not waived by the source |
| 2) | Be accurate | 2. Identify the deceased with the barest information necessary to allow for confirmation of verification of the story, except if privacy is required |
| 3) | Show respect | 3. Show respect for the deceased in how they are described and portrayed and how their death is reported |
| 4) | Avoid vague and sensational language | 4a. Use of neutral language and tone as opposed to sensational language that could exacerbate grief and lead to shock |
| | | 4b. Avoid unnecessary details of the death |
| 5) | Seek out independent experts | 5. Engage independent experts to scrutinise claims and provide a nuanced assessment of the death and events leading to it. |
| 6) | Distinguish advocacy from reporting | 6a. Distinguish clearly between statements of fact and opinion |
| | | 6b. The journalist does not take sides on the content of the article, but is focused solely on providing an accurate, balanced and complete report |
| 7) | Be original | 7. Report firsthand or cite original source |

scored <7 of the maximum 10 points obtainable. We reported the frequency and percentage of articles that met each of the 10 criteria and speaker identity in each country and across all countries. For the qualitative content analysis on how maternal deaths are reported, we used narratology, which examines texts with an emphasis on meaning produced by a piece's structure and choice of words [28]. The reporter's viewpoint, choice of phrase, and the language's tonal quality were examined. These predetermined codes were matched to passages of text in the report, thereby improving the systematicity of our analysis [20].

Two coders (KR and AB-T) independently coded all selected articles using the coding sheet (S2 File). As per best practice, inter-coder reliability for the quantitative content analysis was assessed for agreement between coders using Cohen's kappa (κ) [20]. Following an iterative process with multiple rounds of discussion aiming for consensus, we achieved an average inter-coder reliability (κ) of 0.92 (S2 File), with κ > 0.75 accepted as excellent [29]. For the qualitative content analysis, rather than estimating inter-coder reliability, we implemented a detailed audit trail and debriefing meetings to ensure we could represent the multiple realities of the content in the media through the lens of the two coders [30,31].

In instances where a specific maternal death was reported by more than one news source, we analysed content on 'what about' the maternal death as a unique case, while citing the multiple sources together. However, we analysed 'how' the maternal death was reported as individual reports, citing the multiple sources separately. We chose this divergent approach because the case being reported is one, but how it is reported may vary.

## Ethical considerations

Ethical approval for this study was obtained from the Research and Ethics Committee of the London School of Hygiene & Tropical Medicine (Ref number: 30564). All data were obtained from publicly available sources. However, recognising the sensitivity of the subject matter, no identifying information on the pregnant woman who died, her associates, or those involved in her care was extracted for this study.

## Results

### Characteristics of included articles

A total of 330 articles were retrieved across all five categories. Most of the retrieved articles reported accidental/incidental causes of death (153/330), with many describing pregnant women dying from traffic accidents, violence, or natural phenomena such as floods. This was followed by 84 articles which described maternal health as a broad issue (S1 File). Of all 322 articles, 53 provided a detailed obstetric narrative of specific maternal deaths [32–84] (Fig 1).

Of all 53 included articles, there were 35 (66%) online newspaper articles [33–36,38,39,41,44,45,47–50,53,56,57, 60–70,76,77,79–84], 16 (30%) blogs [32,37,40,43,51,52,54,55,58,59,71–75,78], and two (4%) online investigative reports [42,46]. With the exclusion of two articles in Nigeria that provided detailed obstetric narratives of three maternal deaths

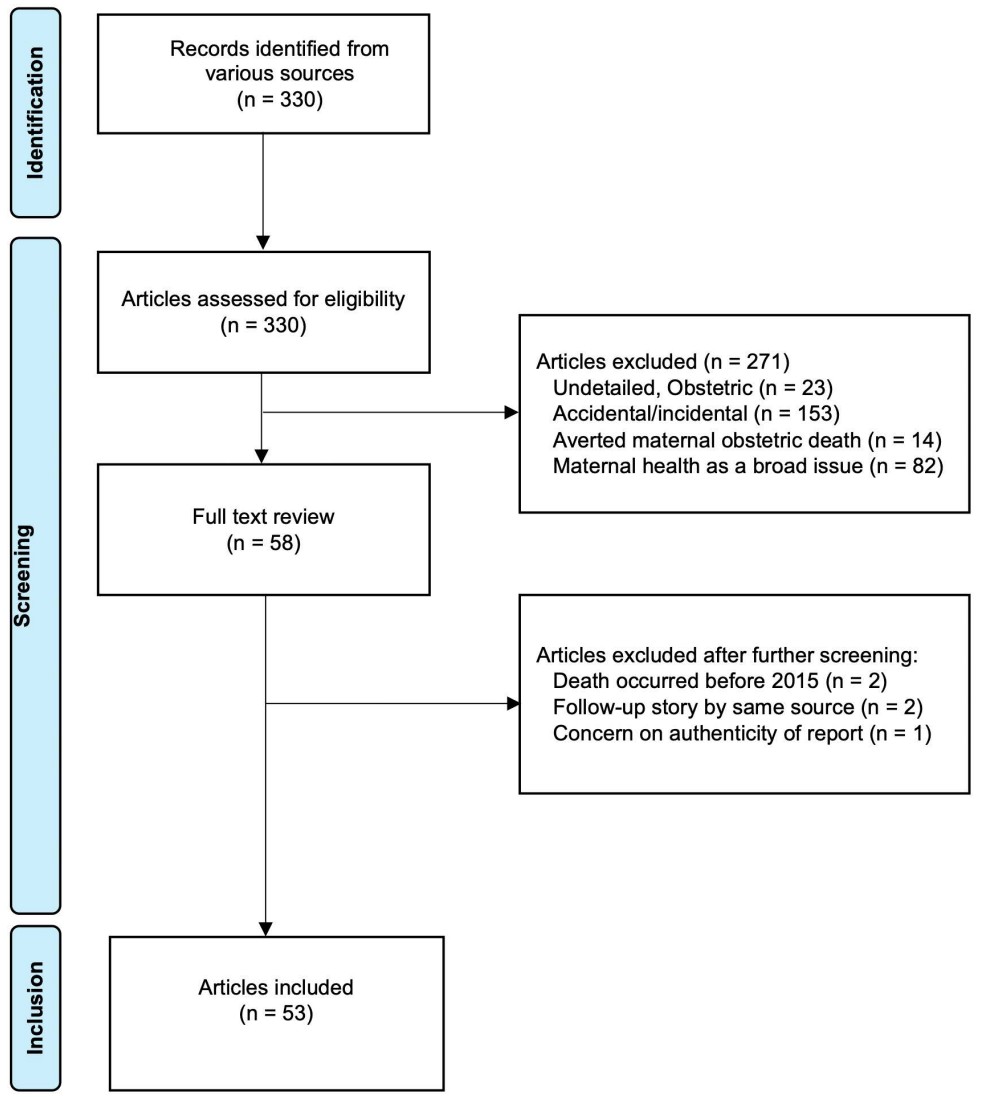

**Fig 1. PRISMA chart detailing article selection.**

[62,77], all other 51 included articles provided detailed obstetric narratives of a specific maternal death [32–61,63–76, 78–84] (S3 File).

Across all articles, Nigeria had 36 unique maternal deaths published across 40 articles on maternal death (75%) [34,36,38–44,46–54,56,58–62,64–66,68–70,73–80,83,84], followed by The Gambia with four unique stories across five articles (9%) [32,35,63,67,82], Liberia with three unique stories in three articles (6%) [33,37,55], Ghana also with three unique stories in three articles (6%) [45,57,81], and Sierra Leonne with two unique stories in two articles (4%) [71,72] (S3 File).

Over the years, two articles were published in 2017 [55,78], three in 2018 [45,57,62], two in 2019 [32,33], 11 in 2020 [38,39,54,58,59,63,66,67,73,76,77], three in 2021 [37,71,82], three in 2022 [35,51,56], 16 in 2023 [34,36,40,41,46,47,49, 50,52,53,59,60,64,65,68,72], nine in 2024 [42–44,48,61,69,70,74,75], and six in 2025 [48,79–81,83,84] (S3 File).

## What about maternal deaths are being reported?

All 53 articles with detailed obstetric-related maternal death reports contained elements aligned with the three-delay framework in some form [32–84]. Of these, issues relating to Phase III delays were mentioned at least once in 45 articles [32–36,42–54,56–65,67–76,78–84], 13 articles described issues related to Phase II delays [33,37–39, 42,48,58,62,66,74,75,77,83], and six articles centred on issues related to Phase I delays [34,40,41,55,62,75] (Table 3 and S3 File).

**Phase I delays.** In Nigeria, two articles reported that the thought of having to travel great distances, as much as 58 km in some parts of the country, to the nearest health facility was a disincentive to deciding to seek care. In one article, this consideration was linked to concerns about the riskiness or affordability of transport options [34,62]. Another article highlighted concerns about affording high hospital fees, including a case in which there was a delay in seeking alternative private care when public hospital doctors in Nigeria were on strike [65]. Another explanation for the delay in seeking care is the poor reputation of health facilities [62,75]. Another article highlighted that a desire to avoid non-vaginal birth contributed to the delay in seeking formal care [40]. Three articles noted the influence of a woman's social network on the decision to seek care [34,62,75]. Specifically, some pregnant women who died were not allowed to or were strongly discouraged from seeking skilled care by close or influential family members [40,62]. In convincing pregnant women not to seek care, there was some comparison made with other women. For example, it was reported that one pregnant woman who died was told that she should be "*delivered of her baby at home like many other women*" [34].

In Nigeria and Liberia, religious reasons were given as reasons for delaying health-seeking in the articles retrieved [41,55]. The article from Liberia reported that concern with long distance and pressure from family members sometimes forced women to seek traditional birth attendants (TBAs) local to them, some of whom believe women have complications because they were involved in adultery [55] (Table 3 and S3 File).

**Phase II delays.** Across The Gambia, Nigeria, and Liberia, the most reported issues with physically reaching appropriate care were external factors, such as bad roads [33,37,55,62,75] and distance to functional facilities [33,37,55,62,75]. Two articles from Nigeria highlighted the weather as a contributory factor to the delay in reaching a health facility [75,77]. Also, in Nigeria, the death of a pregnant woman was attributed to a delay caused by being stopped by police en route to a facility while they were trying to enforce COVID-19 protocols during the lockdown [39,58,66], and another was delayed while the police tried to ascertain if she was being kidnapped [38]. In one instance, a police man was alleged to have assaulted a pregnant woman who later died by hitting her with a torchlight on the head [38] (Table 3 and S3 File).

For direct travel to facilities, one article from Nigeria reported showed a husband accompanying his semi-conscious pregnant woman in a car travelling to a health facility [48], while other articles reported that due to lack of safer options in some communities, motorcycles were commonly used to travel to facilities in an emergency, despite the riskiness involved [62,75]. In one article, it was reported that a pregnant woman who had to take a motorcycle to reach care was involved

**Table 3. Delays and exemplar quotes from published articles.**

| Type of delays reported in phases | Newspaper (Country) | Exemplar narratives |
|---|---|---|
| **Phase I delay** | | |
| Cost of care | *Daily Trust (Nigeria)* | "…they decided to stay back due to their inability to afford the medical bill at the private hospitals" [65]. |
| Distance to travel disincentive | *The Punch (Nigeria)* | "[the husband] would never have thought of risking the life of his six-month-old pregnant wife, carrying her on a motorcycle on a distance of 25km, but for the unavailability of good health care facilities in the community" [62]. |
| Expensive transportation | *International Centre for Investigative Journalism (Nigeria)* | "A single motorcycle ride can cost N1500, an amount many families simply cannot afford" [75]. |
| Society and culture | *Linda Ikeji's Blog (Nigeria)* | "Client was offered an [elective (pre-planned) caesarean] because she has a heart condition. Her frail heart would simply not withstand the distress of labour and vaginal delivery! Her reply: I need to tell hubby about all this first, and she left… Never showed up until weeks after. This time, in labour; but the baby's heart had stopped beating... When confronted, husband was like: I had an agreement with God that my wife will deliver normally… Long story short, the lady is in the mortuary. Her heart gave up - a cardiac arrest" [40]. |
| Bad hospital reputation | *The Punch (Nigeria)* | "We just come here for the antenatal classes only, nothing more. When it's time for delivery, some of us don't come here… There is no reputable hospital here" [62]. |
| Religious beliefs | *National Accord (Nigeria)* | "The pregnant woman has come under heavy criticisms… for refusing to undergo CS because her husband "*had an agreement with God that she will deliver normally.*"" |
| **Phase II delay** | | |
| Weather | *International Centre for Investigative Journalism (Nigeria)* | "During dry season the road is wet and muddy and further delay travel time" [75]. |
| Police delays | *Premium Times (Nigeria)* | "The police stopped the motorcycle to question why the lockdown order was not being obeyed. They were delayed for hours while [she] continued to bleed" [39]. |
| Road accidents | *The Punch (Nigeria)* | "Barely 15 minutes as the couple rode on the motorcycle on the tarred road leading to Ejigbo, an oncoming car, while overtaking an 18-seater bus, colluded with the [family's] motorcycle, knocking them down" [62]. |
| No transport available, so alternative locally available care sought | *Front Page Africa (Liberia)* | "… Resolving to take his wife to the clinic which is distance away across the river became the last resort for [him]. But canoes do not come by easily. He had to wait with his wife who was already in labour for the return of the canoe from across the river. Alas, [she] did not end the journey alive. And so was the baby" [55]. |
| Inter-facility delays | *News Public Trust (Liberia) Buzz capital (Nigeria)* | "… upon their arrival at the Kolahun hospital, there was no doctor at the facility and that they were asked to provide $7,000LD ($US35) for the purchasing of fuel for the hospital ambulance to do another referral to the Tellewoyan Hospital in Voinjama City" [33]. "...the hospital had no provision for ambulance emergency transport; it was just as if we were in a dead trap" [74]. |
| Bad roads | *Front Page Africa (Liberia)* | "The hospital's ambulance cannot ply the road owing to the deep cuts and damaged bridges… [A pregnant woman] died… all because of the bad road" [37]. |
| **Phase III** | | |
| Long waiting time | *The Punch (Nigeria)* | "[The woman] was allegedly left in pain for more than eight hours without attention from the medical personnel on duty" [68]. |
| Unavailability of staff | *The Nation (Nigeria)* | "…there was no doctor on ground to attend to them, except an already exhausted doctor on call who was visibly overwhelmed by the enormity of the challenges before him" [70]. |
| Alleged staff incompetence | *Buzz capital (Nigeria)* | "Surgeon was not competent, my wife received multiple colloids and crystalloids but huge delay in starting surgery increased their mortality risk significantly. Surgeon could not secure multiple bleeders at any point he lacked the surgical skills to handle the situation, my wife and son literally bled to death" [74]. |
| Staff withholding care | *Front Page Africa (Liberia)* | "The midwife backed off the delivery, superstitiously accusing Martha of committing adultery for which she was suffering complications and having prolonged labour" [55]. |
| Care withheld for payment | *Daily Guide Network (Ghana)* | "…met her untimely death in what her family claims is a case of negligence by doctors at the health facility, as she was denied service after failing to pay GH¢500 as being a motivation fee for the surgeon" [45]. |

in a road traffic accident en route before she died of obstetric causes in the facility. There was one article that reported that women some had to walk to facilities during the emergency [62]. In the case of a woman from Liberia, one article described the use of canoes, which they had to wait for to cross the river to reach land before travelling by road to a facility [55]. Delay in inter-facility transfer after they have reached a facility was also mentioned as being contributory to death in Liberia and Nigeria. Underpinning this contributory factor were reasons such as health facilities lacking a functional ambulance or families being told to pay for the fuel needed for the facility's ambulance to transfer them, which they could not afford [33,42,74,83]. Though in some instances an ambulance was available and used to transfer, journeys were prolonged because of bad roads, and some women still died along the way or upon reaching the health facility. Women were reported to have died in transit either because of the transport delay or the worsening of their complication while travelling to a health facility in emergency [37,55,61,62] (Table 3 and S3 File).

**Phase III delays.** Once at the hospital, the most common issue reported relating to the death of pregnant women was long waiting time to receive care due to staff shortage or unavailability of skilled health personnel (SHP) at the time of the emergency. This was widely reported in Nigeria and in one unique case each, in The Gambia and Ghana [42,45,46,51–54,59,60,63–65,67,68,74,75,84]. Some issues raised in Nigeria related to alleged negligence, poor management, or malpractice by SHP [35,42,46–48,51,52,54,56,57,59,61–64,68,70,72,74,78,83,84]. An example of these issues was recounted by a husband of a pregnant woman who died in Nigeria who said, he "*wept uncontrollably for the gruesome way [his] wife and son were killed out of gross negligence*," as the staff were apparently "*incompetent with no clue on how to perform [cardiopulmonary resuscitation] for fresh still birth,"* leaving him to perform it himself [42]. In The Gambia, there was a reported case in which care was delayed because of concerns that the woman might have had COVID-19 [63,67] while in Liberia, a midwife withheld care to a pregnant woman in labour while accusing her of committing adultery [55]. Other pregnant women who died had their care delayed because it was contingent on payment, referred to as "*motivation fees"* in Ghana [45] or "*deposit"* in Nigeria [48], online payment problems [52,53,64], and issues with obtaining new currency from banks to pay for care after policy change in Nigeria [50,68]. Relatives described how they struggled to pay the bills and secure the blood requested by SHP for women who passed [34,36,46,50,52,53,64,65,68]. In one article from Northern Nigeria, it was reported that the SHP on duty did not accept the bank transfer that had been done by the family of the woman, as the facility policy was cash-in-hand only [80].

Across all countries, other Phase III issues reported included factors such as a lack of medication and supplies [32,42,46,62,63,74,75] and infrastructural issues including lack of space, electricity or pipe-borne water to provide needed care to pregnant women [62,63]. In Nigeria, it was reported that the SHP needed to use their mobile phones as flashlights to see the operating site when electrical power was out [42,74]. In another case from Nigeria, a female doctor en route a facility to attend to an emergency was held up by the police [54,73,76] because they assumed she was a "*prostitute, because a doctor would never dress the way she dressed"* [54] (Table 3 and S3 File).

### How are maternal deaths being reported?

Of all included articles, 20 were assessed as high quality [33,35,46,50,52,53,55,56,61,63–65,67,68,75,78–80,83,84], 27 were assessed as average quality [32,34,36–39,42–45,48,49,51,54,57–62,69,71,74,76,77,81,82], and six articles were assessed as poor quality [40,41,47,70,72,73] (S4 File).

**Assessment of reporting standards.** *Recognise that most stories involve a degree of nuance*: Thirty-six articles included more than one voice (68%) [32–36,38,39,41,46,48–50,52–58,60–70,75–77,79,80,84]. With 21 articles [34–36,42–45,47,51,56,64,74,83,84], the voice of spouses of the deceased, [34–36,42–45,47,51,56,64,74,83,84] was the most captured, followed closely by the government cited in 20 articles [35,38,39,49,50,52,56,58–60,62,64,66,68,72,75–77,79], health facility management in nine articles [33,45,52,53,63,67,71,80,83], and SHP in seven articles [40,41,46,55,62,70,75]. There were also three articles that cited human rights advocates [33,38,39] and two that cited unaffiliated people online who were not experts in the field [32,63]. Of all articles, 42 named information sources were

consulted in the article [32–38,40–59,61–69,71,73–77] with *"unconfirmed"* and *"anonymous"* sources cited in five articles [38,39,54,63,70] (Table 4 and S4 File).

In many articles from Nigeria, statements attributed to spouses and other relatives mostly described the events preceding the death and in many, blamed either SHP, government or the health system for the pregnant woman's death [42,51,53,64,74,78,79,83,84].

Also, in several articles published in Nigeria and in one article from Liberia, when SHP were engaged, some of their statements were about denying allegations of misconduct, blaming the relatives of the pregnant woman for her death or highlighting the pregnant woman's contributions to her death [33,40,46,53,62,70,83]. For example, one doctor discussed how the pregnant woman and her baby both died during birth because the husband insisted on vaginal delivery over the recommended caesarean section due to a pre-existing heart condition that put the pregnant woman at higher risk of complications in labour [40]. Another doctor explained that they needed to get food and rest after working 24 hours, and that was why they were away from their station when an emergency arrived at their facility [70].

There were a handful of instances when the voices of the government described intended actions following the maternal death. For example, government representatives announced investigations into deaths in Ghana, Nigeria, and Sierra Leone [35,45,52,68]. In Nigeria, other reported planned actions were shutting down a facility [56] and recognising the need to construct new and rehabilitate existing primary health centres while acknowledging limited funds [62]. In one Nigerian unique case reported in two articles, a State Governor and a government representative in the legislature expressed their regret and sympathies following protests in the community because of the maternal death [59,60]. In another article from Nigeria, it was reported that the legislature paid an unscheduled visit to inspect the facility where the death had occurred [79]. However, other contributions from the government in Nigeria focused on denying culpability of SHP [64] or police [38,58,66] in maternal deaths. In The Gambia, the main referral facility launched an inquiry following the death of a pregnant woman in an ambulance while she was waiting to receive care, though in the same article, an official of the facility was already making a case of 'no fault' of the SHP in the death [63,67]. In another case from The Gambia, the government was awaiting a report from the hospital to determine the next course of action [35]. In Ghana, it was reported that an audit on why the death occurred was *"expected to be carried out within seven days in line with administrative procedure after informing the metropolitan and regional directorates of the incident"* [45]. Similarly, one article in Liberia included advocates calling for a government-led inquiry [33]. In Sierra Leone, it was reported that a midwife had been placed under

**Table 4. Details about sources by country.**

| Speaker Identity in relation to pregnant woman | Nigeria | Liberia | Sierra Leone | Ghana | The Gambia | Total |
|---|---|---|---|---|---|---|
| Family | | | | | | |
| Husband | 18 [34,36,42–44,47–53,56,62,64,74,83,84] | | | 2 [45,57] | 1 [35] | 19 |
| Mother | 1 [34] | | | | | 1 |
| Sibling | 3 [46,47,62] | | | 1 [57] | | 3 |
| Unspecified | 4 [42,49,74,75] | 1 [33] | | 1 [45] | | 5 |
| Extended Family | 1 [39] | | | | | 1 |
| Government official | 16 [50,52,56,58–60,62,64,66,68,75–77] | 1 [33] | 1 [72] | | 1 [35] | 16 |
| Healthcare worker | | | | | | |
| Doctor/Nurse/Midwife | 6 [40,41,46,62,70,75] | 1 [55] | | | | 7 |
| Health facility management | 2 [52,53,80,83] | 1 [33] | 1 [71] | 1 [45] | 2 [63,67] | 9 |
| Traditional Birth Attendant | 1 [77] | | | | | 1 |
| Unaffiliated observer of the incident | 11 [38,41,46,48,53,54,61,70,73,76,77] | 1 [55] | | | | 12 |
| Other | 11 [38,39,46,60–62,66,68–70,75,77] | 2 [37,55] | | | 3 [32,35,63] | 16 |
| Different news source | 3 [54,58,73] | | 1 [71] | | 1 [32] | 5 |

investigation, and elsewhere the Ministry of Health had suspended over 15 SHP for negligence [71,72] (S3 File and S4 File).

Among those articles, almost all from Nigeria, that used unnamed sources to provide context for maternal deaths [38,39,54,63,70], two included statements expressly stating that the informant was speaking under the condition of anonymity [38,63]. However, in two others, *"the source"* described events leading to the maternal death but was not identified, nor was a request for anonymity established [39,54]. In another, *"unconfirmed sources"* were used to explain the possible role of favouritism in the rota allocation of doctors in a hospital, which meant some doctors were not allocated shifts and were not penalised for not coming to work and to discuss the lack of hiring new medical staff [70]. One blogger in Nigeria repeatedly cited social media posts as the source of the information for reporting about the maternal deaths in their blog posts [40,43,51].

**Provide information to aid verification**: Of all included articles, 47 articles provided information sufficient to identify the pregnant woman who died in cases in which privacy was not stated as having been requested [32–36,38,39,42–65,67–69,72–84]. In the six articles that did not, four were from Nigeria and reported on three maternal deaths [40,41,66,70] while the remaining two included one each from Liberia [37] and Sierra Leone [71] (S4 File).

**Show respect**: Forty-eight articles showed respect in how they described and portrayed the pregnant woman who died, avoiding graphic pictures of death and non-dignifying or derogatory words [32–38,40–61,63–71,73–84] (S4 File). Of the remaining five, one blog article from Nigeria contained graphic photos after the woman's death, including one photo of her visible under a clear tarp, and in another photo, the face of the dead baby could be seen, as he/she was surrounded by many onlookers [59]. In two other articles, also from Nigeria, a pregnant woman who had died was shown placed on a surface in a room [39,58]. In one published in Sierra Leone, a pregnant woman was addressed as a *"destitute"* [72].

**Avoid vague and sensational language**: Thirty-seven articles, spread across all included countries, were assessed to have used neutral language in their headline and full narrative [33,35,37–41,43–46,50,52,53,55,56,59–61,63–69,71,72,74,75,77–80,83,84]. The other 14 articles, mostly from Nigeria, were deemed to have used some form of sensational language [32,34,36,42,47–49,51,54,57,62,70,73,76] (S3 File). For example, one article by *The Nation* described a general hospital as having the "*semblance of a graveyard*" [70]. To convey the pain of losing a man's pregnant wife, the author described the recurring memory of his wife passing "*like a musical track put on a repeat*" [62].

**Avoid unnecessary detail**: Fifty-two articles avoided unnecessary detail in narrating maternal deaths [32–61,63–84] (S4 File). Of these, three articles that took an investigative journalistic approach provided detailed descriptions of the medical issues leading up to the death [42,46,74]. For example, one article described how the wife "*…kept frequently using the washroom to urinate and felt urges to pass stools. At about 5 a.m., a repeat of Vaginal misoprostol was done, but they claimed she was not still contracting*" [74]. However, one article published in The Gambia contained details on the work that a pregnant woman was doing on a farm the day before the death occurred, a narrative that did not appear necessary for the story [62].

**Seek out independent experts**: Three articles, all published in Nigeria, included consultation with an independent expert in the article [62,75,84] (S4 File). For example, one article in *Punch Nigeria* includes a statement by a Lagos-based gynaecologist who, in addressing the unsafe practice of pregnant women taking motorcycles to the hospital from a local community, stated that "*the situation puts the unborn babies in danger should an accident occur, and the tummy is hit. It's too risky. Jarring the baby around may not be the wisest choice. The risk of an accident on a motorcycle is so much greater than being in a car*." The expert gynaecologist also encouraged local government to take action, stating "*they can provide public transport for the women*" [62].

**Distinguish between advocacy and reporting**: All articles distinguished between statements of fact and opinion, and authors were not seen to take a side on the content of the article [32–84] (S4 File).

**Be original**: Of the 46 articles included, there were 10 sets of articles reporting the same maternal death (Set A [40,41], Set B [43,44], Set C [47,50], Set D [59,60], Set E [61,69], Set F [63,67], Set G [73,76], Set H [42,74], Set I [52,53,64], and

Set J [38,39,58,66]. All sets reporting the same death were from Nigeria, except Set F from The Gambia (S5 File). For the most part, these articles covered the same maternal death but varied in terms of content and perspective. However, there were a few instances in which the exact same article was republished without attribution to the source. For example, Daily Trust's "*Pregnant Woman Dies Over Naira Scarcity*" published 12th February 2023 [50] made no attribution to the original article in the Tribune titled, "*A pregnant woman loses her life to scarcity of new naira notes*" published 2nd February 2023, though it clearly builds on the content published in the earlier article [47]. In another set, the same maternal death was being reported, yet two articles described her as ~name withheld" and "yet to be identified" [38,66], while the other two articles named the woman [39,58]. Further, an article by *The Point* published 4th September 2020 titled, "*Woman dies after EFSTH nurses allegedly failed to admit her*", includes a more detailed description of the events leading up to the death, such as the woman being referred to multiple different health centres due to displaying COVID-19 symptoms, before being attended to by a doctor and passing away soon after [63]. Another article published on the same day by *The Standard* titled, "*EFSTH Accused of Neglecting Patient to Die*", covers the same death but includes different details surrounding the mother's death and additional perspectives, including those that blamed the hospital heavily for their misconduct [67] (S4 File and S5 File).

## Discussion

The objective of this study was to examine what is reported about maternal deaths in Anglophone West Africa and how these deaths are reported in digital media. Across countries, we identified 53 articles that met our inclusion criteria, covering 48 unique reports of maternal deaths. Nigeria alone accounted for 75% of the retrieved articles, covering 36 (75%) unique reports. Overall, 66% of articles were published as online newspaper articles, and 30% as blog posts. The most frequently reported factors that led to maternal death in the included articles were closely aligned with issues related to receiving appropriate care at healthcare facilities, i.e., Phase III of the three-delay framework, which were mentioned at least once in 85% of the articles. Among these, specific contributory factors such as SHP negligence, long waiting times, staff unavailability, malpractice, and withholding care until payment were most frequently mentioned. All articles distinguished fact from opinion, and authors avoided taking a side, 90% were respectful, 68% captured more than one voice, with most articles capturing the voices of the spouse and the government; however, only three (6%) consulted independent experts.

### Interpretation of results

According to our analysis, the majority of the articles identified were published in Nigeria. This is not entirely surprising, as the annual number of maternal deaths occurring in Nigeria (75,000 in 2023) is significantly higher than estimates from the other Anglophone countries in West Africa (a cumulative of 4,410 in 2023) [1]. The severity of a country's maternal mortality may influence how the media in that country reports the public health issue. While online newspapers constitute the predominant source of detailed reports on maternal deaths in Anglophone West Africa, blogs have increasingly emerged as a formidable platform for maternal death narratives, especially since the COVID-19 pandemic, with examples retrieved from Liberia, Nigeria, and Sierra Leone. This increase in blogging popularity aligns with observations from a global survey of bloggers, which found that they blogged more and experienced higher traffic during the pandemic [85]. In our study, more than 80% of 16 blog articles that provided detailed reports of maternal deaths came from one blogging site in Nigeria alone, *Linda Ikeji's Blog*. According to the blog's owner, the blog's initial purpose was "*gossip*" and "*entertainment*," but it has since morphed to include "*news*" [86]. In this case, it means that an expectation of comprehensive and proportional health news coverage, as defined by the AHCJ standards, cannot be perceived as overly ambitious [25,26]. For online investigative reports, we only retrieved two [42,46]. We reckon this might reflect the challenges faced by investigative journalism across many parts of SSA, including a lack of funding, government influence, media censorship, insecurity, threats and lawsuits, and poor access to information [87–89].

In terms of what about maternal mortality was reported, results of our media content analysis found that delays within Phase III were the most frequently reported as contributing to maternal death across most Anglophone West African countries. Contrarily, the few published MPDSR reports published in these countries have mostly pointed to other phases as being the most common contributions to maternal deaths. For example, two facility-based MPDSR-based studies led by obstetricians and conducted in Nigeria reported that amongst women who made it to health facilities before dying, delayed presentation in health facilities in the first place, which includes a combination of Phase I and II delays, contributed to most maternal deaths [90,91]. In a community-based MPDSR in Liberia, contributory factors to phase I delays were most frequently cited, followed closely by those in phase III delays [92]. This disagreement with the published literature might point to systematic variation in conclusions on contributory delays depending on who is asking (Media vs MPDSR Committee members, who are usually SHPs themselves), or perchance the media's preference for reporting more salacious stories. It is probably more *"interesting"* to report that a maternal death occurred because of negligence of an SHP than because a woman delayed her travel to a health facility.

While the relative distribution of attributable delay leading to maternal death may differ between the media and community-based MPDSR reports, there is no dispute among researchers that Phase III delays remain a health system challenge requiring urgent action, especially in Nigeria [91], where we found several reports of such delays. Awaiting fee before service, incompetence, long waiting times, lack of medicines and supplies, infrastructural issues, and staff shortages were all issues highlighted in media reports. These issues are not dissimilar to those reported in the literature regarding obstetric service provision in the country [93–96]. The underpinning drivers of these issues point to a lack of training, poor health system regulation, and, recently in the country, an exodus of SHP known locally in the country as the *"jápa syndrome"* [84,97,98]. However, the issues we unpacked from the media reports are not limited to Phase III delays or to Nigeria; many have also been highlighted in the academic literature. For example, difficulties with travel to care for some women in Nigeria, which point to recognised issues with Phase II delays [99–101] and limited supplies and materials such as oxygen in Bong County, Liberia [92], reveal that other countries have issues that need to be addressed also. There are also issues reported in the literature that we did not find in the included media reports. For example, a lack of knowledge of danger signs has been reported to cause delay in seeking care in The Gambia [102] and referral in Liberia [103]. There were issues we identified from media reports that have not been reported in the literature, and that might not have been captured in traditional data sources such as an MPDSR report [21,104]. For example, a doctor being held up travelling to reach the hospital because a police officer stopped them en route thinking they were a prostitute [54,73], inclement weather, violence, ambulances that could not be used to transfer women because they could not travel on bad roads or patients could not afford to fuel them [42,74], non-provision of care due to failure to pay *"motivation fees"* [45], online payment alert delays [52,53,64], and issues with non-acceptance of Nigeria's old currency at a time when there were issues in obtaining the new notes from banks [50,68].

Regarding how maternal deaths are being reported, it appears to be a case of ping-ponging blame, especially in Nigeria, with husbands blaming health facility staff for the demise, while SHP, when engaged, absolved themselves of any responsibility and proceeded to blame husbands. Most articles that reported maternal deaths included the voice of government representatives for the most part, followed by the husbands of the deceased. It was not clear from our findings why the government's voice was widely captured in the included articles across countries, or why, when captured, the government felt the need to release statements to defend SHP or the police in Nigeria. The role of the government includes providing accessible and affordable, high-quality health services, enforcing laws to protect people from violence and disrespectful care, and supporting the population in making healthy choices [105,106]. To perform this role, fairness and accountability are crucial. As such, governments defending SHP without awaiting findings from inquiries or reports from maternal death audits undermine their capacity to function effectively as stewards of the health system. There was also a disproportionate lack of patient-facing SHP perspectives included, despite Phase III delays being mentioned in 85% of articles. Though it appears that facility management tended to speak on behalf of the facility rather than having SHPs

speak directly to the media. We found examples of this being done in all five countries [33,45,52,53,63,67,71,80,83]. This might point to recognition of possible legal implications of SHP testimony for potential inquiries, legal challenges, and medical malpractice investigations, with awareness of rights to litigation reported to be increasing in many Anglophone West African countries [83,107,108].

There was also the practice of anonymous sources being engaged to highlight contributory factors to the maternal death relating to the health facility or government inadequacy. There was a striking lack of independent expert voices, which were consulted in only 6% of the articles we retrieved across all Anglophone West African countries. According to AHCJ, experts provide the balance needed to effectively communicate complex health issues, such as is the case for many maternal deaths, to the public [25,26]. Almost all articles avoided unnecessary details about the death; however, some still left gaps that prevented readers from getting a complete account of the events leading to the death. Also, while we retrieved three articles (two blogs and one online newspaper) in Nigeria that included posthumous images of the deceased [39,58,59], a practice considered disrespectful [27], this was not the predominant practice. The media in Nigeria has been called out for having a habit of picturing corpses of dead people [109]. However, the use of sensational language and a persuasive tone was an issue in a third of the articles, mainly blogs, which is probably a reflection of the underlying writing style of blogs aimed at clickbaiting potential readers.

## Strengths and limitations

This study is a pioneering effort to provide a broad understanding of what is reported and how maternal deaths are reported in the print media across Anglophone West Africa. The media content analysis used in the study allows for the reporting of maternal deaths that occur outside health facilities, which are known to be under-reported in these settings. It also allows for capturing multiple perspectives on the events leading to maternal deaths. We double-coded to minimise bias and missingness in data extraction from the included articles and held numerous team discussions to resolve disagreements at each stage of the research process. The application of piloted and defined eligibility criteria enhanced comparability across articles, while testing inter-coder reliability improved the reliability and validity of the analysis.

However, there are limitations to keep in mind in interpreting our results. First, despite our best efforts, there were only a few articles with detailed accounts of maternal death articles in Ghana, Liberia, Sierra Leone and The Gambia compared to the high number retrieved from Nigeria. This made it difficult for us to draw comparisons in content and reporting quality across countries, as we had intended. While this finding is most likely a reflection of reality and represents the relatively lower burden of maternal deaths in these other Anglophone West African countries compared to Nigeria, it is also plausible that we could have missed some published articles. However, we made every effort to maximise our search results, including launching a social media campaign and asking peers and colleagues in maternal health working in the countries to share any relevant articles they may know of. Second, we only included articles that were published online, publicly available at the time we conducted our search, and published by the top four to five newspaper outlets and the two most popular blogging sites per country. This, therefore, means we might have missed some articles. Also, the majority of the articles we retrieved were published from 2020 onwards, despite our search for articles from 2015. We opine that this recency bias might have more to do with the archiving of older articles by online news sources than with trends over the period of interest. As such, it is possible that articles were published that were not available when we searched. However, we searched repeatedly throughout the study period to minimise the risk of missing such articles. Third, we included only English-speaking West African countries in this study, even though English, French, and Portuguese are spoken across West Africa. While we recognise that this might have excluded some relevant articles, even in the included countries, we made this choice because of the differential effect of language on reporting style and content [18,110]. By restricting our analysis to Anglophone countries, we have endeavoured to hold the overarching media system and journalistic traditions influenced by language relatively constant, allowing for a more precise and undistorted analysis of how localised sociocultural, institutional, and editorial factors specifically influence the framing and reporting of maternal

deaths. We encourage future research to conduct similar analyses in other West African countries, including Francophone and Lusophone countries. Fourth, in our study, we included only reports of maternal deaths following pregnancy-related complications, as defined by the WHO. However, we also retrieved over 150 reports on pregnancy-associated maternal deaths from all Anglophone West African countries resulting from incidental/accidental causes including gender-based violence, conflict, murder for baby, rituals, police brutality, and road traffic accident, including for some pregnant women while in transit to a health facility [111–117]. Such reports offer insights into other critical exposures pregnant women face that require attention. Fifth, we included articles with 250 words or more. While this approach might have introduced some bias into our findings, our review of the less detailed articles presented in S1 File suggests that, although their inclusion would have increased the number of eligible articles, the overall conclusions would not have changed. Finally, we adapted the AHCJ framework [26] and the Impress Standards Code [27], both of which have been used in other studies to analyse media articles across the Global North [118,119], but not in settings like the countries in our study. We recognise that cultural preferences might influence media practices, and as a result, our adaptation emphasised quality while also prioritising cultural adaptability.

## Implications for policy and practice

Remarkably, our findings from analysing digital media reporting of maternal death align with some of the epidemiological trends in maternal mortality. For example, our analysis aligned with the empirical evidence that the highest number of maternal deaths occurs in Nigeria and there was an increase in excess maternal deaths in the year of the COVID-19 pandemic [1,120,121]. This may lend some credence to the utility of insights from media content analysis for tracking and monitoring public health priorities. Similar utility was realised from a previous media content analysis which tracked and reported health and social consequences of the COVID-19 pandemic in West Africa [122] and another that examined newspaper coverage of progress made towards realising the Millennium Development Goal 5 on reducing maternal mortality in two sub-Saharan African countries, Rwanda and South Africa [15]. Our findings show that additional insights and perspectives emerged, which should be considered valuable for MPDSR efforts in countries. It is well documented that most of the information captured for MPDSR comes from facility case notes. However, case notes are written up by SHP, and most of the details they contain regarding events leading to the death only start from the arrival of the woman at the facility [9,104]. MPDSR Committees need to consider securing additional information from the media to complement data captured from facility case notes and interviews to support fairer, more comprehensive reviews that will inform policy.

Also, based on the findings of the study, the high proportion of Phase III delays suggests that improvements are needed in the health workforce and infrastructure across all five countries to improve outcomes for pregnant women. Training has been shown to be effective in improving capacity of SHPs to provide the care that pregnant women need in emergencies, and such training, if well implemented, guarantees value for money [123,124]. This will potentially address some of the competence issues reported in the media. In addition, minimising financial and geographical barriers to care for women in emergency who reach health facilities, while ensuring the availability of quality care, needs to be a top priority for governments, especially in Nigeria. Delayed travel to care increases odds of maternal death [125,126], and cost is a well-evidenced barrier to care access, more so the emergency obstetric and newborn care package that pregnant women require at the time they face an obstetric emergency that could lead to death [127]. User fee exemption policies, which have been well received by stakeholders broadly and shown to be effective in low-resource settings such as Anglophone West African countries, should be considered [128–130]. There is also the need to improve obstetric referral decision-making and communication [131]. In Nigeria, where we found a number of police encounters occurred before some maternal deaths, cultural reorientation of the police and other security operatives is needed. Such effort will be particularly crucial for a country where the public perception of the police is at best, mixed [132].

However, before insights from media reportage can be fully maximised, a greater understanding by those in the media, including new media such as blogs, of their role and expectations is needed. The media should present facts, not gossip,

hearsay, or conjecture. Even a recent TV reportage on a maternal death in Nigeria shows that this understanding might be lacking, with the reporter asking, *"Who should 'we' believe?"* after presenting social media quotes on the maternal death [133]. 'We' should be the population waiting for accurate news from the media. There needs to be an improvement in quality of news reports which discuss obstetric-related maternal deaths in detail, so that women and their communities, as well as policymakers, have a better understanding of the risks that pregnant women face. All articles, and not just most, as we found in our study, should maintain respect and dignity of pregnant women even in death by avoiding graphic images and derogatory names. These standards could be achieved by mandating refresher courses for journalists to familiarise or refamiliarise them with the AHCJ principles, or with a new, commonly agreed-upon standard of health reporting culturally relevant to the countries. Additionally, given the popularity of many blogs, bloggers should be required to take a similar course before they can report on important topics such as maternal death. In reporting maternal deaths, focus should be on presenting facts and not opinions, providing details, capturing perspectives of independent experts along with other voices, including SHP who provided care, treating the dead with respect and dignity, and minimising blaming [134]. We argue that these standards should apply to all deaths involving pregnant women, not only those resulting from complications of pregnancy and childbirth, as we identified a few articles that included undignified posthumous images of pregnant women involved in accidents [39,58,59]. Nomenclature used to refer to the pregnant woman who died should be respectful and cultural appropriate, as expected in verbal autopsies [135]. Furthermore, as misinformation is rife on social media [136], authors, particularly those who write blogs, need to minimise dependence on social media posts to put together stories of maternal deaths, as was seen in some articles included in our study. In addition, greater emphasis on actions taken by the government to investigate the contributory factors in the death, offer sympathy to the bereaved, and, more broadly, address prevailing maternal health challenges within the health system, as was done in a handful of articles [56,77,137–139], will be helpful. Finally, recognising the potential for comprehensive reporting as observed in our study, the limited engagement of investigative journalists in reporting on the complex issue of maternal mortality should be addressed. Non-government actors working to reduce maternal mortality can be crucial partners in supporting, informing, and equipping investigative journalists to publish reports of maternal deaths that can trigger change [140].

## Conclusion

Our findings indicate that media reportage of maternal deaths in Anglophone West Africa through digital platforms both corroborates existing evidence and provides new insights into the causes of maternal mortality in the region. While there are some notable good practices in how maternal deaths are being reported in the digital media across many Anglophone West African countries, there remains a need for universally applied standards, including for new media such as blogs. Priority standards include ensuring that the dignity and respect of pregnant women are preserved, even in death; engaging multiple voices, including independent experts, for comprehensive, balanced, accurate, and informative reporting; and clearly reporting on actions being undertaken or required to prevent future occurrences. Standardising these practices would maximise the media's potential to contribute to efforts to reduce maternal mortality in the countries and in the region at large.

## Supporting information

**S1 File. News source selection.**
(DOCX)

**S2 File. Coding framework and results of intercoder reliability testing.**
(DOCX)

**S3 File. Data extraction file.**
(XLSX)

**S4 File. Assessment of how matwrnal deaths were reported.**
(XLSX)

**S5 File. Report duplication analysis.**
(XLSX)

## Acknowledgments

This project was undertaken as part of the fulfilment of the requirements for the MSc in Public Health at the London School of Hygiene and Tropical Medicine. We are grateful to colleagues and several experts in the field from the included countries who provided insights on which news sources to include. Special appreciation to Dr Uduak Okomo, Dr Mardieh Dennis, Dr Cephas Avoka, Mrs Chisom Obi-Jeff, Dr Abimbola Olaniran, and Dr Michael Ezeanochie.

## Author contributions

**Conceptualization:** Aduragbemi Banke-Thomas.

**Data curation:** Kiran Roy, Noimot Balogun, Aduragbemi Banke-Thomas.

**Formal analysis:** Kiran Roy, Noimot Balogun, Aduragbemi Banke-Thomas.

**Investigation:** Kiran Roy, Aduragbemi Banke-Thomas.

**Methodology:** Kiran Roy, Aduragbemi Banke-Thomas.

**Project administration:** Aduragbemi Banke-Thomas.

**Resources:** Aduragbemi Banke-Thomas.

**Software:** Aduragbemi Banke-Thomas.

**Supervision:** Aduragbemi Banke-Thomas.

**Validation:** Aduragbemi Banke-Thomas.

**Visualization:** Aduragbemi Banke-Thomas.

**Writing – original draft:** Kiran Roy, Aduragbemi Banke-Thomas.

**Writing – review & editing:** Kiran Roy, Noimot Balogun, Aduragbemi Banke-Thomas.

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
