## [Decision Letter · Decision Letter 0]

18 Jan 2026

PGPH-D-25-01664

Reporting maternal deaths in Anglophone West Africa: A Media Content Analysis

Dear Dr. Banke-Thomas,

Thank you for submitting your manuscript to PLOS Global Public Health. After careful consideration, we feel that it has merit but does not fully meet PLOS Global Public Health’s publication criteria as it currently stands. Therefore, we invite you to submit a revised version of the manuscript that addresses the points raised during the review process.

We look forward to receiving your revised manuscript.

Kind regards,

Emmanuel Olamijuwon, Ph.D

Academic Editor

Journal Requirements:

1. Please ensure that your Ethics Statement is available in its entirety at the beginning of your Methods section, under a subheading 'Ethics Statement'.

2. Please upload separate figure files in .tif or .eps format. Also, remove the figures from your manuscript file but keep the legends.

Additional Editor Comments (if provided):

Reviewers' comments:

Reviewer's Responses to Questions

**Comments to the Author**

1. Does this manuscript meet PLOS Global Public Health’s publication criteria? Is the manuscript technically sound, and do the data support the conclusions? The manuscript must describe methodologically and ethically rigorous research with conclusions that are appropriately drawn based on the data presented.? Is the manuscript technically sound, and do the data support the conclusions? The manuscript must describe methodologically and ethically rigorous research with conclusions that are appropriately drawn based on the data presented.

Reviewer #1: Yes

Reviewer #2: Partly

Reviewer #3: Yes

2. Has the statistical analysis been performed appropriately and rigorously?

Reviewer #1: N/A

Reviewer #2: No

Reviewer #3: I don't know

3. Have the authors made all data underlying the findings in their manuscript fully available (please refer to the Data Availability Statement at the start of the manuscript PDF file)?

The PLOS Data policy requires authors to make all data underlying the findings described in their manuscript fully available without restriction, with rare exception. The data should be provided as part of the manuscript or its supporting information, or deposited to a public repository. For example, in addition to summary statistics, the data points behind means, medians and variance measures should be available. If there are restrictions on publicly sharing data—e.g. participant privacy or use of data from a third party—those must be specified.requires authors to make all data underlying the findings described in their manuscript fully available without restriction, with rare exception. The data should be provided as part of the manuscript or its supporting information, or deposited to a public repository. For example, in addition to summary statistics, the data points behind means, medians and variance measures should be available. If there are restrictions on publicly sharing data—e.g. participant privacy or use of data from a third party—those must be specified.

Reviewer #1: Yes

Reviewer #2: Yes

Reviewer #3: No

4. Is the manuscript presented in an intelligible fashion and written in standard English?

Reviewer #1: Yes

Reviewer #2: Yes

Reviewer #3: Yes

Reviewer #1: 1. Does the manuscript meet PLOS Global Public Health’s publication criteria? Is the manuscript technically sound, and do the data support the conclusions?

The manuscript addresses an important and underexplored public health issue: media reporting of maternal deaths in Anglophone West Africa. The topic is relevant, timely, and well aligned with the journal’s scope. The study design, a media content analysis, is appropriate for the research question, and the authors present a substantial body of extracted data. However, several methodological gaps reduce the technical robustness of the manuscript in its current form.

Specifically, key components require strengthening to meet publication criteria:

i). The rationale for selecting Anglophone countries is insufficient and needs deeper justification (Introduction, paragraph 4, lines 35–42). The authors should explain how Anglophone media systems differ from Francophone ones, and why these differences are important for understanding maternal health reporting.

ii). The Macnamara framework is cited but not operationalised, leaving the analytic procedure unclear (Methods, lines 46–48).

iii). News source selection lacks objective justification. Please provide objective metrics (e.g., circulation figures, page-view statistics) and explain why other possible sources were excluded.

iv). The search strategy is narrow and may not capture the full spectrum of relevant media reports.

v). Eligibility criteria, especially requiring the woman’s name and a word count threshold, may introduce bias but are not adequately justified.

vi). Intercoder reliability is not reported, which is essential for both qualitative and quantitative coding.

vii). The qualitative findings read largely as narrative summaries, with insufficient thematic structure.

The conclusions are generally consistent with the descriptive findings presented, but some claims, particularly in the Discussion, are speculative and require either stronger evidence or more cautious framing.

With these issues addressed, the manuscript could provide a valuable contribution; however, substantial revision is required for methodological and interpretive rigour.

2. Has the statistical analysis been performed appropriately and rigorously?

The manuscript does not include formal statistical analysis. The study relies exclusively on simple descriptive frequencies and qualitative coding. While descriptive statistics are appropriate for a media content analysis, two issues compromise rigour:

I). No intercoder agreement statistics (such as Cohen’s kappa or Krippendorff’s alpha) are reported. Given that coders classified articles into delay phases, quality categories, and thematic elements, reliability metrics are essential to ensure coding consistency.

Although the analytical approach does not require complex inferential statistics, the absence of reliability metrics and opaque scoring criteria undermines analytical rigour and should be addressed.

3. Have the authors made all data underlying the findings in their manuscript fully available?

Yes. The authors provide detailed supporting files (S1–S3), which include:

The full list of included articles and extracted variables, coding outcomes, reporting-standard scoring, assessment of language and sensationalism, and Identification of duplicate stories

These files are comprehensive and align with PLOS’s open-data requirements. However, a formal codebook defining the coding categories is not provided, and the absence of intercoder reliability makes it unclear how consistently the coding was applied. While the data themselves are available, the transparency of the coding framework could be improved to enable full reproducibility.

4. Is the manuscript presented in an intelligible fashion and written in standard English?

The manuscript is generally written in clear English. The structure follows a conventional academic presentation, and the argument is easy to follow. However:

i). Some sections, particularly the Results, are overly descriptive and would benefit from clearer organisation into descriptive characteristics and patterns of reporting.

ii). The qualitative results require more thematic clarity.

iii). The Discussion section occasionally adopts an advocacy tone and includes speculative statements that should be supported with citations or reframed cautiously.

Reviewer #2: Summary:

This manuscript addresses an important topic of how maternal deaths are reported in Anglophone West Africa and what this reveals for maternal mortality surveillance and accountability. The study is timely, theoretically grounded, and methodologically innovative in linking media analysis with maternal health frameworks. However, the manuscript would benefit from clearer justification of methodological choices, a balanced presentation of results, stronger critical reflection on background and discussion, and more actionable implications for both public health and media practice.

Introduction:

“Another source of information on various kinds of health-related topics, including maternal deaths, is the Media.”

Reviewer comment:

The term “media” is broad and requires further elaboration. The authors should clearly define what types of media are included (e.g., newspapers, blogs, social media, investigative journalism) and briefly explain how media functions as a source of maternal health information.

“Newspapers, social media, and blogs play a crucial role in how the public understand and perceive public health issues.”

Reviewer comment:

Social media is a heterogeneous category. The authors should specify which platforms are included under “social media” and clarify how these platforms differ from newspapers and blogs in shaping public health perceptions.

“Both methods help improve understanding of the causes of maternal deaths in sub-Saharan Africa, however, implementation has been fraught with numerous challenges including cost associated with routine data collection, blame culture, and so on.”

Reviewer comment:

This statement would benefit from further elaboration. The authors should expand on the key challenges associated with verbal and social autopsy methods (e.g., sustainability, data completeness, blame culture, feasibility in low-resource settings) to better justify the need for alternative data sources such as media reports.

“While a previous article by Gugsa et al. (2015) examined newspaper coverage of maternal health more generally…”

Reviewer comment:

The manuscript would benefit from a clearer articulation of its novelty. The authors should explicitly state how this study differs from and advances prior media-based research, particularly in terms of innovation, geographic focus, or analytical framework.

“Our objective in this study was to conduct a media content analysis to understand what about and how maternal deaths are reported across print and electronic media outlets in Anglophone West Africa.”

Reviewer comment:

What are print and electronic media outlets, what is the importance of these media outlets on maternal health, and how did authors come to have print and electronic media outlets? Authors need to explain this.

Method:

“Various individual factors relating to delay in seeking care ….contribute to maternal deaths in the sub-region .”

Reviewer comment:

This statement appears to rely on limited citation. The authors should draw on a broader body of literature to support claims on the information of individual factors related to delay in seeking care.

“First, in each country, four to five of the country’s leading online newspapers were selected based on circulation numbers, which can indicate popularity.”

Reviewer comment:

The selection criteria for “leading” newspapers require further clarification. The authors should explain how circulation or popularity was determined and provide supporting evidence or references.

“After this campaign, additional sources including additional blogging sites and investigative reporting sites were identified.”

Reviewer comment:

The inclusion of blogs and investigative reporting sites alongside newspapers requires clearer justification. The authors should explain how these sources align with the study’s objectives and analytical framework.

“An initial preliminary search was conducted to test search terms and assess articles retrieved. For each news source, the search terms “pregnant woman” and “maternal mortality” were used within the search tab function of the website.”

Reviewer comment:

The authors should justify the selection of these search terms and clarify whether other relevant terms (e.g., maternal death and postpartum period) were considered.

“Articles with more than 250 words tended to have sufficient detail needed to help us realise the objective of our analysis.”

Reviewer comment:

A stronger justification is needed for the 250-word cutoff. The authors should indicate whether this decision was informed by pilot testing or prior literature and acknowledge potential exclusion of shorter but influential reports.

“A detailed report of maternal death due to obstetric causes…”

Reviewer comment:

The manuscript should clearly define obstetric versus non-obstetric causes of maternal death, with brief examples, to improve clarity for readers.

Results:

“Of all 46 included articles, there were 29 (63%) online newspaper articles [26 29,31,32,34,37,38,40–43,46,49,50,53–63,69,70], 15 (33%) blogs [25,30,33,36,44,45,47,48,51,52,64–68], and two (4%)investigative reports [35,39].”

Reviewer comment:

Given the dominance of newspaper articles and the limited number of investigative reports, the authors should acknowledge this imbalance and discuss its implications for interpretation.

“Across all articles, Nigeria had 31 unique maternal deaths published across 35 articles on maternal death (76%) [27,29,31–37,39–47,49,51–55,57–59,61–63,66–70], followed by The Gambia with three unique stories across four articles (9%) [25,28,56,60], Liberia with three unique stories in three articles (7%) [26,30,48], Ghana with two unique stories in two articles (4%) [38,50], and Sierra Leonne also with two unique stories in two articles (4%) [64,65]”

Reviewer comment:

As the dataset is heavily weighted toward Nigeria, the authors should avoid strong cross-country comparisons and clearly state that findings primarily reflect Nigerian media coverage.

“Of the 46 articles included, there were 10 sets of articles reporting the same maternal death.”

Reviewer comment:

The authors should discuss whether repeated reporting of the same deaths may have limited the diversity of narratives and consider whether the search strategy could be expanded to capture greater variation.

Discussion:

“More than 30% of the world’s maternal deaths occur in this region.”

Reviewer comment:

This statement should be supported with additional or more recent references to strengthen its credibility.

“As per findings of our analysis, majority of the articles identified were published in Nigeria. This is not entirely surprising, as the annual number of maternal deaths occurring in Nigeria (75,000 in 2023) is significantly higher than estimates from the other Anglophone countries in West Africa (Cumulative of 4,410 in 2023) [1].”

Reviewer comment:

This is not a strong justification. The authors need to provide a better rationale for this.

“In our study, more than 80% of 16 blog articles that provided detailed reports of maternal deaths came from one blogging site in Nigeria alone, Linda Ikeji’s Blog.”

Reviewer comment:

Since findings from the blogging site are based solely on Nigeria, it would be better to limit the analysis to Nigeria instead of attempting a cross-country comparison.

Conclusion:

“When it comes to issues contributing to maternal health …. within their communities.”

Reviewer comment:

Authors should have discussed the role of the media in the background section instead of addressing it here. Since the authors focused on newspapers, blogs, and reports, they need to explain how these three forms of print media can help reduce maternal deaths and what policies different stakeholders can adopt using these sources.

References:

Reviewer comment:

Authors need to format article numbers and references differently. Otherwise, they may appear similar and confuse the audience.

Reviewer #3: Reviewer Comments

General Comments

Thank you for the opportunity to review this manuscript. The topic is timely and of clear relevance, particularly given the role of media in shaping public discourse and, to some extent, political and economic decision-making. Examining media representations of maternal mortality in West Africa is therefore an important and worthwhile endeavour.

Before addressing the manuscript section by section, I wish to raise an issue related to researcher positionality. The lead author appears to be based outside the region, while the co-authors’ affiliations suggest regional representation. Given the importance of contextual understanding in media and situational analysis—especially when interpreting meanings attributed to text and images—this positionality warrants explicit acknowledgement. Clarifying how contextual expertise and local perspectives informed the analysis would strengthen the manuscript. It is also unclear whether this work originated as a graduate research project developed under the supervision of the co-authors; if so, this could be transparently stated.

The introduction is presented rather lightly and would benefit from greater engagement with the broader literature on media consumption, representation, and framing, particularly in relation to maternal mortality. There are few references that reflect the historical and conceptual complexity of this field. For example, the manuscript would benefit from situating the study within existing research and investment in maternal mortality research and advocacy (MMRA) in the region.

The rationale for selecting the specific countries and sub-region is not clearly articulated. While the introduction emphasises high maternal mortality rates in the countries of focus, it remains unclear whether this was the primary selection criterion, or whether practical considerations—such as the availability of English-language media sources—also influenced the choice. As currently presented, the reader is left to infer this rationale. Additionally, the opening statements are broad and predominantly negatively framed, despite the highly nuanced and heterogeneous nature of maternal mortality trends at regional and sub-regional levels. This framing relies on very limited citation and would benefit from a more balanced and contextualised discussion.

As a general point, I recommend a careful review of the language used throughout the manuscript. At times, the tone is overly casual for an academic article. For example, terms such as “blame culture” and phrases such as “beyond counting the numbers and statistics” are used without sufficient conceptual grounding or citation. Several sentences would benefit from revision to improve sensitivity and clarity, particularly those on page 1, lines 35–39.

Methods

The methods section is generally well presented. However, the content analysis procedure referenced would benefit from a clearer description, even if brief. Specifically, it would be helpful to outline how this protocol was developed, what theoretical or methodological framework underpins it, and how it was applied in the context of this study.

Clarification is also needed regarding the search strategy. Were “pregnant woman” and “maternal mortality” the only search terms used? If so, a justification should be provided, along with a discussion of potential limitations. Additionally, the PRISMA diagram should be explicitly referred to as Figure 1 within the text to guide the reader.

Results

There appears to be considerable repetition between the tables and the narrative text, particularly in the presentation of qualitative findings. I recommend revising this section to reduce redundancy and instead draw the reader’s attention to the key nuances, interpretations, and implications of the quoted material—both individually and collectively.

It is also unclear whether both print and electronic media sources were accessed for data extraction; this should be explicitly stated. The presentation of results using extensive numerical references following individual statements is confusing and detracts from readability, particularly in the quantitative content analysis section. While I appreciate the need to reference source articles, the current approach makes it difficult for the reader to follow the main findings. The authors may wish to consider alternative ways of presenting these references (e.g. grouped citations or table-based cross-referencing).

Finally, the limitations section notes the use of social media to identify relevant articles on maternal mortality. However, no corresponding methodological detail is provided earlier in the manuscript. Please clarify whether social media platforms were used as a source of data, what methods were employed, and what this process yielded.

Finally, the recommendations made towards the improvement of maternal health services, including structural and economic determinants are not well placed within the article as it does not sufficiently address these factors (as per the aims). The recommendations to improve journalism are also not necessary unless there was sufficient focus on the media sources used (Blog versus mainstream digital media) and therefore it is hard to make comment on practice. It would be more useful to clear reiterate the key findings as they were identified (as per the aim/s of the study) and question the role of media, the responsibilities of the narrator and journalist given the current democratization and increasing distrust of media as a source of 'truth". These latter points are just to consider in future revision.

**Do you want your identity to be public for this peer review?** For information about this choice, including consent withdrawal, please see our Privacy Policy..

Reviewer #1: **Yes:** Mandu Stephen EkpenyongMandu Stephen EkpenyongMandu Stephen EkpenyongMandu Stephen Ekpenyong

Reviewer #2: No

Reviewer #3: No

---

## [Decision Letter · Decision Letter 1]

8 Mar 2026

PGPH-D-25-01664R1

Reporting maternal deaths in Anglophone West Africa: A media content analysis of articles published online between 2015 and 2025

Dear Dr. Banke-Thomas,

Thank you for submitting your manuscript to PLOS Global Public Health. After careful consideration, we feel that it has merit but does not fully meet PLOS Global Public Health’s publication criteria as it currently stands. Therefore, we invite you to submit a revised version of the manuscript that addresses the points raised during the review process.

We look forward to receiving your revised manuscript.

Kind regards,

Emmanuel Olamijuwon, Ph.D

Academic Editor

Journal Requirements:

Additional Editor Comments (if provided):

Reviewers' comments:

Reviewer's Responses to Questions

**Comments to the Author**

Reviewer #1: All comments have been addressed

Reviewer #3: All comments have been addressed

publication criteria? Is the manuscript technically sound, and do the data support the conclusions? The manuscript must describe methodologically and ethically rigorous research with conclusions that are appropriately drawn based on the data presented.? Is the manuscript technically sound, and do the data support the conclusions? The manuscript must describe methodologically and ethically rigorous research with conclusions that are appropriately drawn based on the data presented.

Reviewer #1: Yes

Reviewer #3: Partly

3. Has the statistical analysis been performed appropriately and rigorously?

Reviewer #1: N/A

Reviewer #3: N/A

4. Have the authors made all data underlying the findings in their manuscript fully available (please refer to the Data Availability Statement at the start of the manuscript PDF file)?

The PLOS Data policy requires authors to make all data underlying the findings described in their manuscript fully available without restriction, with rare exception. The data should be provided as part of the manuscript or its supporting information, or deposited to a public repository. For example, in addition to summary statistics, the data points behind means, medians and variance measures should be available. If there are restrictions on publicly sharing data—e.g. participant privacy or use of data from a third party—those must be specified.requires authors to make all data underlying the findings described in their manuscript fully available without restriction, with rare exception. The data should be provided as part of the manuscript or its supporting information, or deposited to a public repository. For example, in addition to summary statistics, the data points behind means, medians and variance measures should be available. If there are restrictions on publicly sharing data—e.g. participant privacy or use of data from a third party—those must be specified.

Reviewer #1: Yes

Reviewer #3: Yes

5. Is the manuscript presented in an intelligible fashion and written in standard English?

Reviewer #1: Yes

Reviewer #3: Yes

Reviewer #1: The authors have made substantial revisions in response to the reviewer comments. The methodological procedures are now clearer, particularly with the inclusion of intercoder reliability and improved explanation of the analytic framework. The Results and Discussion sections are better organised and more cautiously interpreted, which strengthens the overall rigour of the manuscript. The rationale for focusing on Anglophone countries could still be strengthened by briefly clarifying how differences between Anglophone and Francophone media systems may shape patterns of maternal death reporting. Nonetheless, the authors have addressed the major concerns satisfactorily, and I recommend the manuscript for acceptance.

Reviewer #3: The manuscript has been revised and is improved from the first version - all reviewer comments are very well addressed. I do still see scope for additional revision to further improve the grammar and sentence construction before it is ready for publication.

**Do you want your identity to be public for this peer review?** For information about this choice, including consent withdrawal, please see our Privacy Policy..

Reviewer #1: **Yes:** Mandu Stephen EkpenyongMandu Stephen EkpenyongMandu Stephen EkpenyongMandu Stephen Ekpenyong

Reviewer #3: No

---

## [Editor Report · Decision Letter 2]

22 Mar 2026

Reporting maternal deaths in Anglophone West Africa: A media content analysis of articles published online between 2015 and 2025

PGPH-D-25-01664R2

Dear Dr. Banke-Thomas,

We are pleased to inform you that your manuscript 'Reporting maternal deaths in Anglophone West Africa: A media content analysis of articles published online between 2015 and 2025' has been provisionally accepted for publication in PLOS Global Public Health.

Best regards,

Emmanuel Olamijuwon, Ph.D

Academic Editor